# Development of Plum Seed-Derived Carboxymethylcellulose Bioink for 3D Bioprinting

**DOI:** 10.3390/polym15234473

**Published:** 2023-11-21

**Authors:** Juo Lee, Sungmin Lee, Jae Woon Lim, Iksong Byun, Kyoung-Je Jang, Jin-Woo Kim, Jong Hoon Chung, Jungsil Kim, Hoon Seonwoo

**Affiliations:** 1Department of Animal Science & Technology, College of Life Science and Natural Resources, Sunchon National University, Suncheon 57922, Republic of Korea; juolee23@naver.com; 2Interdisciplinary Program in IT-Bio Convergence System, Sunchon National University, Suncheon 57922, Republic of Korea; 3Department of Human Harmonized Robotics, College of Engineering, Sunchon National University, Suncheon 57922, Republic of Korea; 4Department of Biosystems Engineering, College of Agriculture and Life Sciences, Seoul National University, Seoul 08826, Republic of Korea; 5Department of Agricultural Machinery Engineering, College of Life Science and Natural Resources, Sunchon National University, Suncheon 57922, Republic of Korea; 6Department of Bio-Systems Engineering, Institute of Smart Farm, Gyeongsang National University, Jinju 52828, Republic of Korea; 7Institute of Agriculture & Life Science, Gyeongsang National University, Jinju 52828, Republic of Korea; 8Department of Biological & Agricultural Engineering, University of Arkansas, Fayetteville, AR 72701, USA; 9Materials Science & Engineering Program, University of Arkansas, Fayetteville, AR 72701, USA; 10ELBIO Inc., Seoul 08812, Republic of Korea; 11Department of Bio-Industrial Machinery Engineering, College of Agriculture and Life Sciences, Kyungpook National University, Daegu 41566, Republic of Korea; 12Department of Convergent Biosystems Engineering, College of Life Science and Natural Resources, Sunchon National University, Suncheon 57922, Republic of Korea

**Keywords:** carboxymethyl cellulose, plum seed-derived carboxymethyl cellulose, bioprinting, bioink

## Abstract

Three-dimensional bioprinting represents an innovative platform for fabricating intricate, three-dimensional (3D) tissue structures that closely resemble natural tissues. The development of hybrid bioinks is an actionable strategy for integrating desirable characteristics of components. In this study, cellulose recovered from plum seed was processed to synthesize carboxymethyl cellulose (CMC) for 3D bioprinting. The plum seeds were initially subjected to α-cellulose recovery, followed by the synthesis and characterization of plum seed-derived carboxymethyl cellulose (PCMC). Then, hybrid bioinks composed of PCMC and sodium alginate were fabricated, and their suitability for extrusion-based bioprinting was explored. The PCMC bioinks exhibit a remarkable shear-thinning property, enabling effortless extrusion through the nozzle and maintaining excellent initial shape fidelity. This bioink was then used to print muscle-mimetic 3D structures containing C2C12 cells. Subsequently, the cytotoxicity of PCMC was evaluated at different concentrations to determine the maximum acceptable concentration. As a result, cytotoxicity was not observed in hydrogels containing a suitable concentration of PCMC. Cell viability was also evaluated after printing PCMC-containing bioinks, and it was observed that the bioprinting process caused minimal damage to the cells. This suggests that PCMC/alginate hybrid bioink can be used as a very attractive material for bioprinting applications.

## 1. Introduction

Three-dimensional (3D) printing, commonly referred to as additive manufacturing, indicates the process of incrementally constructing 3D objects one layer at a time [1,2,3]. In this field, 3D bioprinting represents a specialized technique that involves the precise dispensing of bioinks to create intricate 3D tissue structures, enabling the faithful replication of essential cellular interactions observed in vivo. These interactions, often absent in traditional two-dimensional (2D) in vitro cell culture systems, promote cellular behavior that closely mimics the in vivo environment [4,5,6,7]. A variety of bioprinting methods are available, such as extrusion-based, laser-assisted, light-induced, and inkjet-based bioprinting [8,9]. Among these methods, extrusion-based bioprinting is highly versatile and widely used due to its ability to work with many different bioinks, ease of use, and cost-effectiveness [3,10,11]. Hydrogels are a highly promising bioink category based primarily on their ability to create highly hydrated, biocompatible 3D environments conducive to cell growth and proliferation [12,13]. In extrusion-based bioprinting, a bioink must exhibit three essential properties [14,15]. Firstly, it must demonstrate shear-thinning behavior, allowing it to be easily extruded through a pressurized nozzle while still maintaining its form [16,17]. Secondly, it must possess adequate mechanical strength to support the 3D structure that is being printed [18]. Finally, it must shield the enclosed cells from mechanical stresses that may arise during printing [19]. For these purposes, synthetic polymer-based bioinks offer excellent mechanical properties, stability, and tunable characteristics. However, their non-biodegradable nature raises concerns about biocompatibility and environmental impact [20,21]. In contrast, natural polymer-based bioinks provide advantages such as biocompatibility and biodegradability, making them suitable for 3D bioprinting applications [22]. However, natural polymer-based materials may demonstrate inferior mechanical properties and stability relative to synthetic polymers [23]. Consequently, the creation of natural polymer-based materials that possess heightened mechanical properties is essential. This offers the opportunity to surpass the limitations linked to traditional natural polymers [24].

High-molecular-weight natural polymers such as alginate [25], collagen [26], hyaluronic acid [27], and chitosan [28] have gained popularity in 3D printing bioinks due to their printability, biocompatibility, and biodegradability. Cellulose offers numerous advantages as a bioink material [29,30]. The cellulose bioink is a sustainable material choice due to its natural sourcing and superior biodegradability, effectively reducing its negative environmental impact [31]. Additionally, the versatile nature of cellulosic bioinks allows for easy manipulation of their properties, which is particularly advantageous for applications in 3D printing and various manufacturing processes [32,33]. Recycling cellulose, a renewable raw material for bioinks, from discarded resources, offers significant benefits, including sustainability through resource conservation, cost efficiency, waste reduction, and reduced environmental impact [34]. The benefits of using cellulose obtained from fruit seeds as a bioink are manifold, with noteworthy repercussions in the spheres of environment, economy, and pharmaceuticals [35]. Using fruit seed waste enables resource recycling and reduces waste disposal, exemplifying an environmentally friendly and sustainable material practice [36]. Additionally, cellulose, a naturally derived substance, is both inherently safe and exhibits a high degree of biocompatibility, making it invaluable for applications related to biomaterials and drug delivery systems [37,38]. Therefore, this bioink serves as a versatile biomaterial that can be easily adapted to various fabrication and 3D printing processes, offering economic efficiency and versatility over conventional materials [39,40]. These advantages emphasize the potential benefits of incorporating fruit seed-derived cellulose as a bioink [41]. Collectively, these qualities establish cellulose as a highly promising option for bioink [42]. Plum is commonly used in various food products, such as plum extract, in the Republic of Korea, but the disposal of their seeds after consumption has become a significant environmental concern [43,44]. As a result, there is a pressing need for research that focuses on upcycling these plum seeds into valuable products.

In this study, cellulose derived from discarded plum seeds (PC) was recovered and used to synthesize carboxymethyl cellulose (PCMC). The resulting material was developed as a material for bioink (Figure 1). Sodium alginate was incorporated into PCMC hydrogel to improve its rheological properties. Similar to natural polymer-based bioinks like cellulose, sodium alginate is frequently used as a rheological modifier to enhance the viscoelastic characteristics of bioinks. In detail, plum seeds that were discarded after consumption were collected, and the cellulose present in the seeds was recovered. The synthesis of carboxymethyl cellulose (CMC) involves the alkalization of cellulose, followed by its carboxymethylation utilizing chloroacetic acid. A physicochemical analysis was conducted on the recovered and synthesized cellulose and carboxymethyl cellulose. The chemical properties of the cellulose and carboxymethyl cellulose synthesized in our study exhibited similar characteristics to their commercially available counterparts. Rheological properties, such as shear-thinning behavior, extrusion performance, and shape accuracy, were evaluated. Subsequently, a relevant bioink formulation was investigated to print a muscle-mimetic 3D structure with C2C12 cells. The development of bioink using carboxymethyl cellulose derived from discarded plum seeds has suggested that it is possible to establish a sustainable industrial ecosystem through resource recycling and solve the carbon recycling issue. Additionally, the use of discarded plum seeds is expected to contribute to preventing environmental pollution.

## 2. Materials and Methods

### 2.1. Materials

All analytical chemicals, media components, and reagents used included the following substances: sodium hydroxide (415413, Sigma Aldrich, St. Louis, MO, USA), chloroacetic acid (C19627, Sigma Aldrich, St. Louis, MO, USA), sodium carboxymethyl cellulose (419281, Sigma Aldrich, St. Louis, MO, USA), cellulose (310697, Sigma Aldrich, St. Louis, MO, USA), acetic acid (A0051, SAMCHUN, Seoul, Republic of Korea), sodium alginate (7528-1405, DAEJUNG, Busan, Republic of Korea), calcium chloride anhydrous (2507-1400, DAEJUNG, Busan, Republic of Korea), formic acid (F1358, DAEJUNG, Busan, Republic of Korea), hydrogen peroxide (4158-4400, DAEJUNG, Busan, Republic of Korea), ethanol (4204-4410, DAEJUNG, Busan, Republic of Korea), Dulbecco’s Modified Eagle’s Medium (DMEM, LM 001-05, WELGENE, Gyeongsan, Republic of Korea), Dulbecco’s Phosphate-buffered saline (DPBS, LB 001-02, WELGENE, Gyeongsan, Republic of Korea), fetal bovine serum (FBS, S101-07, WELGENE, Gyeongsan, Republic of Korea), penicillin-streptomycin (15140122, Gibco, Grand Island, NY, USA), dimethyl sulfoxide (DMSO, D2650, Sigma Aldrich, St. Louis, MO, USA), water soluble tetrazolium salt (WST-1) assay kit (EZ-3000, DoGenBio, Seoul, Republic of Korea), LIVE/DEAD assay kit (L3224, Invitrogen, Carlsbad, CA, USA), and C2C12 (CRL-1772, ATCC, Manassas, VA, USA).

### 2.2. Recovery of Cellulose

Plum seeds were kindly provided by a farm in Suncheon, Republic of Korea. The plum seed samples were thoroughly washed with tap water to remove pulp, epiphytes, and debris, and dried to a constant weight at a temperature of 50 °C. Next, the seeds were ground and placed into a flask. An organic acid solution was prepared by blending formic acid with acetic acid in a 7:3 ratio. The mixture of 85% formic acid and acetic acid was combined with plum seeds and organic acid in a 1:8 ratio. It was then boiled for 2 h and left at room temperature. Filtration using filter papers was performed afterward, followed by a wash with 80% formic acid and subsequent rinsing with deionized water at 98 °C. Further processing entailed subjecting the pulp to two more rounds of boiling at 80 °C for 2 h each, employing peroxy-formic acid and peroxyacetic acid. Subsequently, it underwent another filtration and was washed five times with hot deionized water to guarantee a thorough elimination of any remaining chemical substances and lignin (Figure A1).

### 2.3. Synthesis of PCMC

The PCMC was prepared following a method described in the literature [45]. To perform this procedure, 5 g of cellulose obtained from plum seeds was mixed with 30 mL of a 30% NaOH solution and isopropanol in a ratio of 1:5 in a beaker at room temperature for 2 h. After the excess solution was removed, it was mixed with a 40% solution of chloroacetic acid and continuously stirred at 80 °C for 2 h. The resulting solution was stored in an oven at 50 °C for 12 h. After heating, the solution was separated into two layers. The upper liquid layer was discarded, and the lower solid layer was dispersed in 67 mL of methanol. The solution was neutralized using 70% (*v*/*v*) acetic acid and then filtered. Both the characteristic assessment of PCMC and the formulation of PCMC in bioink were adjusted to a pH of 7. To remove unwanted by-products, the resulting product was washed five times with 70% (*v*/*v*) ethanol (200 mL). Finally, the resulting product was oven-dried at 55 °C for 24 h [46].

### 2.4. Characterization of PCMC

X-ray diffraction (XRD) measurements (Shimadzu, XRD-7000, Kyoto, Japan) were performed on PC and PCMC to identify phase constitutions in each powder using Cu Kα radiation in a 2θ range of 4–50° at a scan speed of 4°/min. The chemical structural characteristics of PCMC and PC were analyzed by a Fourier transform infrared (FT-IR) spectrophotometer (PERKIN ELIMER, Spectrum Two, PERKIN ELIMER, Waltham, MA, USA). FT-IR analysis was performed in the wavelength range of 4000 to 500 cm^−1^. Commercial carboxymethyl cellulose and cellulose were used as control groups for the analysis. The degree of substitution (DS) was calculated as the ratio between the intensity at 1605 cm⁻¹, representing the carboxyl group (-COOH) stretching vibration, and the intensity at 2920 cm⁻¹, corresponding to the stretching vibration of methyl groups (C-H). The porosity and microstructure of plum seeds, PC, PCMC, and PCMC/Alg, were evaluated using a scanning electron microscope (SEM, JSM-7610F Plus, JEOL, Tokyo, Japan). Dry samples were mounted on carbon discs and sputter coated with a thin layer of pure gold using a magnetron sputtering reactor equipped with a pure Au target and operated in DC mode with a bias voltage of 150 V (Leybold, Cologne, Germany). The microscope was operated at an accelerating voltage of 5 kV, a working distance (WD) of 5 mm, and a spot current of 98 pA, detecting secondary electrons at 1000× and 5000× magnification scales. The surface of PCMC/Alg was visualized using SEM before and after treatment with CaCl_2_. All samples were adequately dried for analysis at a 1000× magnification scale.

To evaluate the cytotoxicity of PCMC and determine the maximum allowable concentration for inclusion in the bioink as a biomaterial, cytotoxicity evaluations of PCMC were conducted. C2C12 myoblasts were cultured in DMEM supplemented with 10% FBS and 1% penicillin-streptomycin under a 5% CO_2_ atmosphere at 37 °C. The cytotoxicity evaluation was carried out using the water soluble tetrazolium salt (WST-1) assay. In brief, C2C12 cells were seeded into individual wells of a 96-well plate at a density of 5 × 10^4^ cells per well and cultured for one day. Subsequently, the DMEM mixture with varying concentrations of PCMC was used to replace the culture medium. The C2C12 cells were then cultured for another day. Following this, the medium was aspirated, and wells were washed with PBS. A 10% WST reagent-containing medium was added to each well and incubated for 1 h. The resulting formazan product’s absorbance at 450 nm was measured using a spectrophotometer, allowing for the quantification of cell viability.

### 2.5. Bioinks Preparation

PCMC/Alg bioink was composed of PCMC and sodium alginate. The experimental groups were categorized based on the alginate concentration, resulting in hydrogels with 10% PCMC with 2% alginate (PCMC/Alg-2), 3% alginate (PCMC/Alg-3), and 4% alginate (PCMC/Alg-4), respectively. PCMC was stirred at 250 rpm using a magnetic stirrer until complete dissolution in a mixture of distilled water and DMEM at a concentration of 10% (*w*/*v*). At the same time, a PCMC/Alg bioink was created by adding 2, 3, and 4% (*w*/*v*) of sodium alginate to the mixture and vigorously stirring at 500 rpm with a magnetic stirrer. Electrostatic interactions were induced through a syringe mixer to facilitate sequential polymerization and achieve a uniform hydrogel.

### 2.6. Rheological Properties

The rheological properties of the PCMC/Alg bioinks were measured using a controlled stress rheometer (MCR-92, Anton Paar, Graz, Austria) equipped with a parallel plate-type adapter (Anton Paar Measurement Adapter C-PP50/XL, 25 mm diameter). A total of 2 mL of each PCMC/Alg bioink was added to the sample holder at 37 °C and allowed to equilibrate for 1 min. Oscillatory shear measurements were conducted over a stress amplitude range of 0.1 to 1000 Pa at a frequency of 1 Hz. Viscosity measurements were performed at shear rates ranging from 0.01 to 100 s⁻^1^.

The overhang capability test of the bioink is conducted by analyzing the mid-range deflection of the suspended filament to determine material collapse. A previously established research platform was adapted, and a model comprising seven columns with known spacings of 1, 2, 3, 4, 5, and 6 mm was created using Autodesk Inventor 2023 (Autodesk, San Francisco, CA, USA) [47]. The platform was fabricated with a Polyjet 3D printer (Object30, Stratasys). Various configurations of PCMC/Alg filaments were deposited onto this platform. Photographs of the deposited filaments were taken immediately after suspension to prevent unwanted material bias. The critical parameters utilized in this test encompassed the feed rate, printing speed, and nozzle diameter, which were adjusted to 120 mm/s, 7 mm/s, and 420 μm, respectively. The reduction area coefficient is determined as the ratio of the area remaining after the filament has collapsed from the theoretical square area to the theoretical square area [47].
(1)$$C_{f}\;\left(\%\right)=\frac{A_{a}}{A_{t}}\times 100_{.}$$

To calculate the reduction area coefficient, C_f_ (%) was applied, as shown in Equation (1), where A_t_ was the theoretical square area and A_a_ was the area remaining after the filament had collapsed from the theoretical square.

### 2.7. Three-Dimensional Printing Ability

A custom 3D bioprinter was fabricated for 3D printing PCMC/Alg bioink. The mechanical support components of the 3D bioprinter were printed using Polyjet 3D printing. The 3D printer setup includes essential components such as a controller board, stepper motor drivers, an LCD controller, and an auto-level controller. These electronic components provide interfacing with a variety of sensors and are responsible for issuing commands that control the behavior of the actuators. The Repetier Firmware Configuration Tool (version 1.0.3) was used to effectively operate the printer. This allows for easy adjustment of parameters such as print speed and ink extrusion to achieve desired configurations. The design of all 3D-printed structures was created using the Inventor program and saved as stereolithography (STL) files. These files were then converted into G-code files using the Cura 15.04.6 slicing program, which generates the necessary coordinates and paths for the 3D printer nozzle during printing. A syringe filled with PCMC/Alg bioink was attached to the 3D printer head. The heating bed temperature was set to 37 °C before the extrusion of the PCMC/Alg bioinks. For printing, a 22 gauge needle with an inner diameter of 420 μm was used. The layer height for printing was 420 μm. The dispensing feed rate was set to 120 mm/min, and the print speed was set to 7 mm/s to match the unique properties of each PCMC/Alg bioink. To evaluate the printing precision for each PCMC/Alg bioink, the printing ability was assessed by printing scaffold support structures with defined porosity. Rectangular grids of 20 × 20 mm were printed, and internal mesh sizes were designed as squares with side lengths of 2, 2.5, and 3 mm, respectively. The measured sizes of the actual printed meshes were then expressed as a percentage of the designed mesh size, providing an assessment of the printing accuracy.

### 2.8. Cell Viability

C2C12 is a mouse myoblast cell line that is used in research to study the differentiation of muscle cells and related biological processes. To fabricate scaffolds loaded with C2C12 cells, it was incorporated into PCMC/Alg bioink at a cell concentration of 2 × 10^6^ cells/mL and then dispensed using a sterile syringe mixer. This mixture was mechanically extruded through a nozzle with an inner diameter of 420 μm. For bioprinting, print speed and extrusion volume were determined using the same method as that used for the printability test. Scaffolds with dimensions of 50 × 50 × 1.0 mm containing embedded cells were bioprinted in 3D culture dishes. The fabricated scaffold was cross-linked with a 2% CaCl_2_ solution for mechanical stability. Subsequently, the scaffolds were rinsed briefly with PBS. After removing PBS by suction, they were gently treated with DMEM, and the media were exchanged every two days with fresh media. To assess cell viability within the PCMC/Alg bioink, a LIVE/DEAD assay was performed on days 1, 4, and 7 following bioprinting. In brief, a staining solution was prepared by adding 5 µL calcein AM and 20 µL ethidium homodimer-1 to 10 mL PBS. After DMEM was removed from the bioprinted scaffolds, 500 μL of staining solution was added directly to the scaffolds and incubated for 30 min. Images were captured using a Leica DM3000 LED fluorescence microscope, and Fiji 2.15 (ImageJ) software was used to count cells and quantify cell viability based on green-stained (live) cells and red-stained (dead) cells.

### 2.9. Statistical Analysis

All the data are presented as a mean ± standard deviation. To evaluate the statistical significance of differences in viscosity, filament fusion, collapse tests, Young’s, and reduced modulus, the distribution of values is considered to be normal. At a significance level of *p* ≤ 0.05, one-way analysis of variance (ANOVA) and Duncan’s new multiple range test were performed to determine the statistically significant differences.

## 3. Results and Discussion

### 3.1. Characterization of PCMC

SEM imaging was performed to observe the surface changes between plum seeds and PC and PCMC. In the case of ground purified plum seed, the surface showed a structure with cellulose grains tightly clustered, and the inherent pores of plum seeds were observed (Figure 2A). On the other hand, on the surface of PC, the natural pores found in plum seeds disappeared, and the size of the tightly packed grains decreased. In the case of PCMC, most of the densely clustered grains were not visible, and spaces between the grains were observed. As a result, the cellulose microstructure of PCMC gradually became smaller in size and acquired a structure that could be easily dissolved or dispersed in solvents.

Crystalline differences between PC and PCMC were evaluated through XRD analysis (Figure 2B). Commercially available cellulose and CMC were also assessed for comparison with plum seed-derived cellulose and PCMC. Typically, the differences in XRD data between cellulose and CMC manifest as characteristic peak shifts, changes in peak intensity and height, and the emergence of new peaks. Carboxymethylation leads to alterations in the intra-molecular crystalline structure, resulting in observable structural differences in the XRD patterns. Compared to regular cellulose, the XRD results for PC exhibited typical peaks (22.4° and 15.5° corresponding to the 002 and 101 planes, respectively) but with reduced crystallinity. PCMC exhibited distinct peaks compared to PC, indicating peak shifts due to carboxymethyl group synthesis. PC and PCMC demonstrated similar peaks to commercial cellulose and CMC, but the presence of peaks at 32° and 45° suggests that trace impurities may still be present in the CMC compound.

FT-IR spectra of PC typically exhibit characteristic peaks associated with the known features of cellulose’s chemical structure (Figure 2C). These peaks are indicative of the chemical composition of cellulose. The peaks observed in PC and PCMC exhibited a similar peak pattern to commercially available ones. In the range of approximately 2850–2900 cm^−1^, a peak is observed, which corresponds to C-H stretching related to methyl and methylene bonds. In the range of 1050–1150 cm^−1^, there is a stretching peak associated with C-O-C (carboxylic ether) bonds, representing one of the cellulose’s distinctive structural features. Additionally, bending peaks related to C-H bonds appear in the range of 1300–1450 cm^−1^. These distinctive peaks confirm that PC is similar in chemical structure to conventional cellulose. The chemical differences between cellulose and CMC are evident in the FT-IR spectra, particularly in the wavelength regions corresponding to hydroxyl groups at 3450 cm^−1^ and carboxyl groups at 1600 cm^−1^ [48]. These pronounced peaks in hydroxyl and carboxyl groups suggest the successful carboxymethylation reaction in both plum seed-derived cellulose and PCMC. This comparison provides insights into the structural characteristics of the PCMC. This indicates that PC and PCMC are essentially no different from the commonly used plant cellulose and CMC, making them suitable for a wide variety of applications. The properties of PCMC are significantly influenced by the degree of substitution. Therefore, for elucidating the relationship between the structure, properties, and applications of PCMC, the calculation of the degree of substitution is essential. CMC is typically known to have a degree of substitution ranging from 0.4 to 1.8. Using infrared spectroscopy, the ratio between the intensity at 1605 cm⁻^1^, corresponding to the stretching vibration of carboxyl groups (-COOH), and the intensity at 2920 cm⁻^1^, corresponding to the stretching vibration of methyl groups (C-H), was calculated, resulting in a DS of 0.98. Typically, it is well-established that when the degree of substitution (DS) is around 0.6–0.7, the emulsification performance is excellent. It is widely recognized that as DS increases, acid resistance and salt tolerance properties are significantly improved.

To assess the maximum tolerable concentration before adjusting the ratio of PCMC in the bioink, cytotoxicity evaluations were conducted at various PCMC concentrations (Figure 2D). PCMC was treated at concentrations of 0.1, 1, 10, and 15% (*w/v*) in DMEM. DMEM without PCMC and DMEM with 5% DMSO were used as negative and positive controls, respectively. Toxicity was lower in the 0.1% and 1% PCMC groups compared to treatment with regular DMEM, while the 10% PCMC group experienced toxicity levels similar to DMEM. However, significantly higher cytotoxicity was observed as the PCMC concentration increased. Therefore, a PCMC concentration of 10%, which maximizes viscosity while maintaining adequate cell compatibility, was selected as the concentration for the bioink formulation.

### 3.2. Characterization of PCMC/Alg Bioink

To investigate the influence of alginate concentration on the microstructure of PCMC/Alg bioinks, SEM analysis was conducted on bioinks treated with CaCl_2_ at various alginate concentrations. The surface of the bioink containing 2% alginate exhibited incomplete cross-linking when treated with CaCl_2_, resulting in a rough appearance (Figure 3A). In contrast, the bioink containing 3% alginate showed extensive cross-linking with a smooth surface. The bioink containing 4% alginate exhibited an excessive distribution of alginate, which promoted unnecessary cross-linking on the surface. PCMC/Alg-2, with its porous structure due to incomplete cross-linking, is expected to positively affect cell viability by potentially facilitating nutrient and oxygen delivery. However, the lack of complete cross-linking may adversely affect the accuracy of the structure during further printing. On the other hand, PCMC/Alg-4 addresses issues related to precise printing structure and long-term culture stability due to its excess alginate content. However, most of the structure becomes cross-linked upon treatment with CaCl_2_, potentially leading to reduced nutrient and oxygen supply from the medium and subsequently lower cell viability. The PCMC/Alg-3 with a suitably cross-linked structure and highly fine porosity is considered the most ideal concentration, providing a mild environment for cells while maintaining the printing structure.

Shear-thinning is a rheological behavior observed in non-Newtonian fluids wherein the viscosity of the fluid decreases as the shear rate increases. This behavior is often referred to as pseudoplastic and is characterized by its absence of time dependence, such as thixotropy. In hydrogel-based printing, shear-thinning properties play a critical role in achieving accurate print results. Hydrogels must maintain their fluid properties as they pass through the print nozzle and quickly recover their shape after extrusion. Therefore, the rheological properties of hydrogels based on the bioink formulation were assessed quantitatively. These measurements provide valuable information about the viscoelastic behavior of the material and its ability to undergo shear-thinning and subsequent recovery. To evaluate the viscoelastic properties of the PCMC/Alg bioink concerning the alginate content, measurements of rheological properties, including storage modulus (G′) and loss modulus (G″), were performed (Figure 3B). In PCMC/Alg-3 and PCMC/Alg-4, it was consistently observed that the storage modulus (G′) exceeded the loss modulus (G″) at all frequencies, indicating that the hydrogel maintained a gel-like state regardless of the applied frequency. However, in PCMC/Alg-2, G′ and G″ exhibited nearly identical values, which, while minimizing shear stress on the cells, might have an adverse impact on printing precision. Furthermore, as the alginate content increased, the complex viscosity showed a corresponding increase. This suggests that a higher alginate content allows the bioink to retain its structural integrity after extrusion, potentially enhancing the mechanical strength of PCMC/Alg bioink. In addition, the shear-thinning behavior of PCMC/Alg bioinks was evaluated, and it was independent of the alginate content (Figure 3C). This property indicates the potential to reduce shear stress during the bioprinting of hybrid hydrogels containing encapsulated cells, ultimately providing a greater ability to minimize shear stress when printing with the hybrid hydrogel-containing cells.

### 3.3. Three-Dimensional Printing Ability

In 3D printing, overhang capability refers to the ability of a printing material to bridge unsupported areas of an object during the printing process. Typically, 3D bioprinting involves depositing ink layer by layer in a vertical direction, with each layer stacking on top of the previous one to build the object. However, overhanging areas of the object have no supporting structure from below and may be susceptible to collapse due to gravity. However, overhanging areas of the object have no supporting structure from below and may be collapsed by gravity. Therefore, the collapse test was evaluated for different alginate concentration compositions of PCMC (Figure 4A). Alginate was added because the hydrogel for bioprinting had insufficient mechanical strength, despite the presence of PCMC, which did contribute to some increase in viscosity. It was observed that an increased ratio of alginate induced minimal deviation in filament deposition. Both PCMC/Alg-3 and PCMC/Alg-4 exhibited impressive overhang capabilities, as they did not collapse between columns situated at varying distances. The collapse area coefficient for each material composition at different distances was determined by dividing the area of the rectangle drawn as the filament passed, assuming a perfectly straight filament, by the area of the rectangle drawn in reality (Figure 4B). Regardless of the alginate content, the collapse area coefficient tended to decrease as the distance between columns increased. PCMC/Alg-2, however, failed to achieve overhangs beyond a 4 mm distance. PCMC/Alg-4 demonstrated the most desirable characteristics, while PCMC/Alg-3 exhibited slightly lower overhang capability compared to PCMC/Alg-4.

To assess the filament fusion and mesh closure effects of PCMC/Alg bioinks, the designed mesh areas of fabricated supports were measured and compared with the intended values (Figure 4C). The printability of each mesh was calculated by dividing the actual mesh size by the designed mesh size. For all groups, there was a trend toward decreasing printability with decreasing mesh size. Qualitatively, PCMC/Alg-4 exhibited better mesh size and geometric structure than the other material compositions (Figure 4D). In the case of PCMC/Alg-2, it was unable to print mesh sizes of 2 × 2 mm and exhibited significantly lower printing ability compared to the other groups. PCMC/Alg-3 exhibited a similar trend in printing ability to PCMC/Alg-4 and was able to print all mesh sizes. This suggests that 3% alginate is the minimum concentration of PCMC/Alg bioink suitable for printing.

### 3.4. Cell Viability

During the deposition of the cell-laden filament through the dispensing nozzle, the cells could potentially experience shear stress, which might be detrimental to cell viability. Therefore, we evaluated the survival rate of cells for a certain period after bioprinting. Live/Dead assays were performed at predefined time points: 1, 4, and 7 days after bioprinting (Figure 5A,B). PCMC/Alg bioinks were cross-linked using CaCl_2_, and a substantial population of both live and dead cells was observed within the structures. On the first day, there were minimal observations of dead cells in PCMC/Alg bioinks, and cell viability was quite similar, with no significant differences. However, the number of dead cells in PCMC/Alg bioinks increased with time, and a trend of decreasing cell viability was observed with increasing alginate content. This is consistent with SEM images suggesting that the presence of alginate reduces porosity within the structure, potentially compromising nutrient delivery to the cells, and with data showing increased shear stress on the cells in rheological evaluations. Therefore, PCMC/Alg-2 and PCMC/Alg-3, with their more porous structures, could provide an increased exchange of necessary growth factors and waste removal while decreasing shear stress, thereby promoting higher cell viability over time.

This novel material source excels in bioink development research due to its environmentally friendly attributes, virtually unlimited material supply, and potential for superior biocompatibility and printability characteristics. It also provides a cost-effective substitute for conventional bioinks, which renders it a multipurpose solution for different bioprinting purposes. However, in food and biomedicine, it is necessary to consider properties such as antibacterial behavior when it involves bioink. When achieving antibacterial properties, it is conventional to mix CMC with various antibacterial agents such as CuO nanocomposites [49], Ag nanoparticles [50], ZnO antibacterial compounds [51], or alternative antibacterial agents to achieve the desired result. By applying these various antibacterial agents, the safety of this bioink can be discussed.

The application of muscle-derived cell-based bioprinting technology has versatile applications in fields such as tissue engineering, regenerative medicine, and biomechanical research. In particular, the incorporation of muscle-derived cells into 3D bioprinting technology offers innovative possibilities for the development of cultured meat. Three-dimensional bioprinting enables more accurate modeling of cellular physiological responses by providing conditions that closely mimic real tissue. These unique features can set it apart from alternative meat and differentiate it in terms of meat production methods and the quality of the meat. Such lab-grown meat production is sustainable, environmentally friendly, and independent of large-scale animal farming practices. Furthermore, the quality and nutritional composition of cultured meat can be finely tuned, offering consumers healthier and more customized food options. This innovation has the potential to revolutionize the meat industry and have a positive impact on the future of food production, providing sustainable, ethical, and health-conscious alternatives to traditional meat products.

## 4. Conclusions

In this study, we developed a hybrid bioink by combining plum seed-derived cellulose with alginate. The chemical properties of recovered cellulose and carboxymethyl cellulose were assessed. The bioink formulations, mixed with alginate at various ratios, exhibited excellent shear-thinning behavior, extrudability, and shape fidelity after deposition. The bioprinting process was found to induce minimal cell death. Overall, the proposed bioink holds great potential for 3D bioprinting applications. PCMC/Alg-2 exhibited beneficial pore formation and excellent cell compatibility with a low alginate content. However, it was observed that its application as a 3D bioprinting ink was affected by low viscosity, resulting in ink collapse and reduced printing precision. In contrast, PCMC/Alg-4 exhibited suitable mechanical strength and printing characteristics for bioprinting, but it had significant disadvantages in terms of pore-clogging and cell compatibility compared to other ink formulations. Therefore, PCMC/Alg-3 is the most suitable bioink for extrusion-based bioprinting due to its favorable rheological properties, precise printability, and adequate cell viability. This research demonstrates that plum seed-derived cellulose can serve as a promising biomaterial in 3D bioprinting processes and highlights the potential value addition of discarded plum seeds. This suggests that agricultural by-products can be used as a material for 3D bioink, providing a possible solution to global challenges such as carbon neutrality.

## Figures and Tables

**Figure 1 polymers-15-04473-f001:**
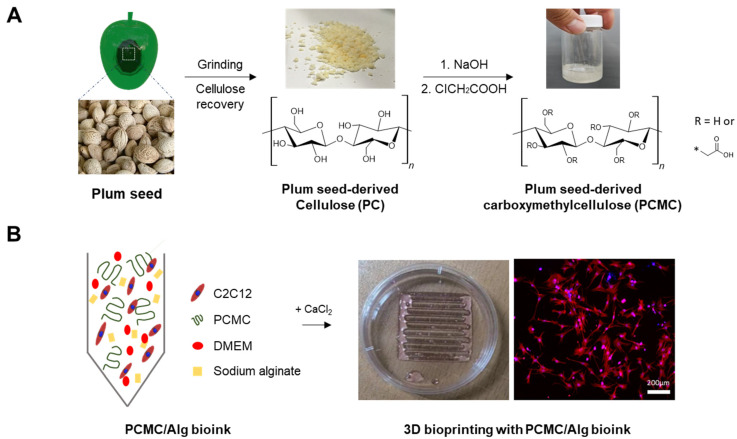
Schematic representation of this study. (**A**) Serial process of plum seed-derived carboxymethylcellulose (PCMC). Cellulose was recovered from finely ground plum seeds, and its characteristics were analyzed through chemical analysis. Carboxymethyl cellulose was synthesized from plum seed-derived cellulose (PC) through an ethylation reaction process. Similar to the procedure for PC, the peaks of carboxymethyl cellulose were verified. (**B**) Schematic illustrating the components of the bioink. The bioink was prepared by dissolving PCMC and sodium alginate in Dulbecco’s Modified Eagle’s Medium (DMEM). Before bioprinting, C2C12 was incorporated into the PCMC/Alg bioink.

**Figure 2 polymers-15-04473-f002:**
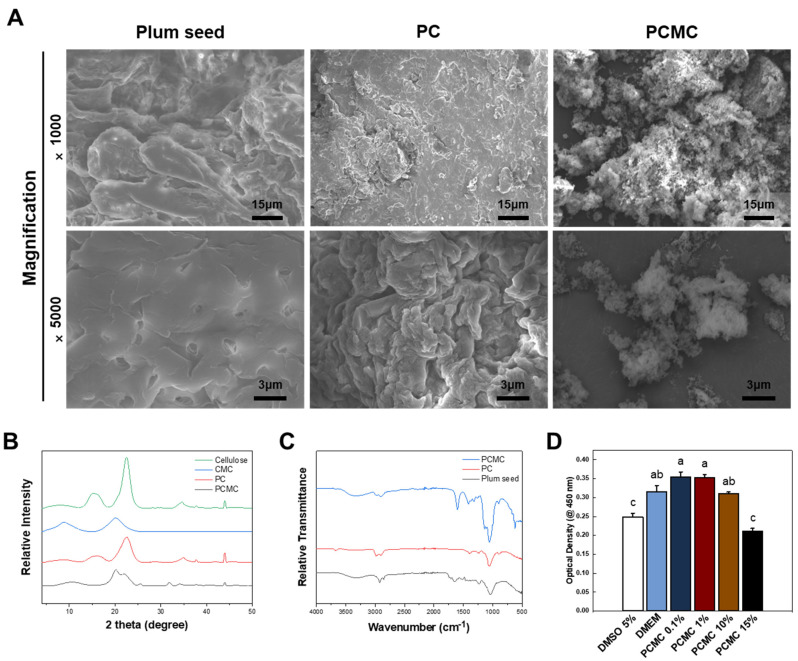
Characterization of the PC and PCMC. (**A**) SEM image displaying the surface of the plum seed, PC, and PCMC. In plum seeds, distinctive pores are observed, and fibers are densely packed. While cellulose does not exhibit these pores, the fibers are still tightly packed. In PCMC, the fibers are dispersed widely and do not exhibit a clustered configuration. (**B**) Crystalline structure of cellulose by X-ray diffraction. The peaks of PC and PCMC appeared to be similar to those of commercial cellulose and CMC, respectively. (**C**) FT-IR spectra of PC and PCMC. These peaks confirm that PC shares similarities with common cellulose in terms of chemical structure. (**D**) WST-1 assay according to the concentration gradient of PCMC. Cytotoxicity evaluations were conducted at various PCMC concentrations to determine the maximum tolerable concentration before adjusting the PCMC ratio as bioink. Error bars indicate the standard mean of errors. The same letter means statistical insignificance (Duncan’s new multiple range test, *p* ≤ 0.05). SEM, scanning electron microscopy; FT-IR, Fourier transform infrared; WST, water soluble tetrazolium.

**Figure 3 polymers-15-04473-f003:**
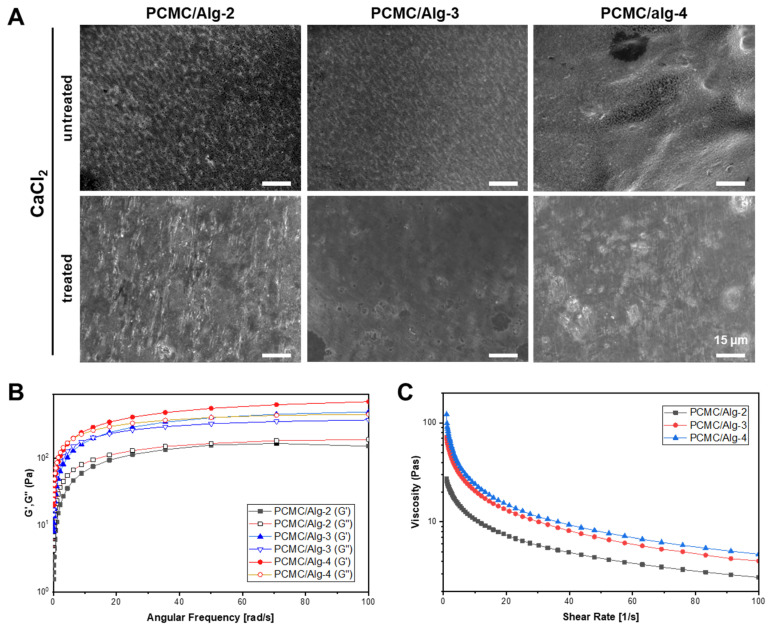
Characterization of PCMC/Alg bioink. (**A**) Morphological analysis of PCMC/Alg bioink by SEM. A comparison was made between images before and after cross-linking using CaCl_2_, and the surface of PCMC/Alg was evaluated based on the alginate content. PCMC/Alg-2 and PCMC/Alg-3 exhibited a consistent distribution of alginate. After cross-linking with CaCl_2_, both retained residual porosity. In contrast, PCMC/Alg-4 showed an excess distribution of alginate on the surface, resulting in complete pore closure as an outcome of the cross-linking process. (**B**) Evaluation of the elastic properties of PCMC/Alg. Storage modulus (G′) and loss modulus (G″) were evaluated as a function of alginate content. (**C**) Evaluation of shear-thinning behavior. PCMC/Alg exhibited characteristics similar to those of hydrogels. PCMC/Alg bioinks exhibited shear-thinning behavior, expecting reduced shear stress during bioprinting with encapsulated cells.

**Figure 4 polymers-15-04473-f004:**
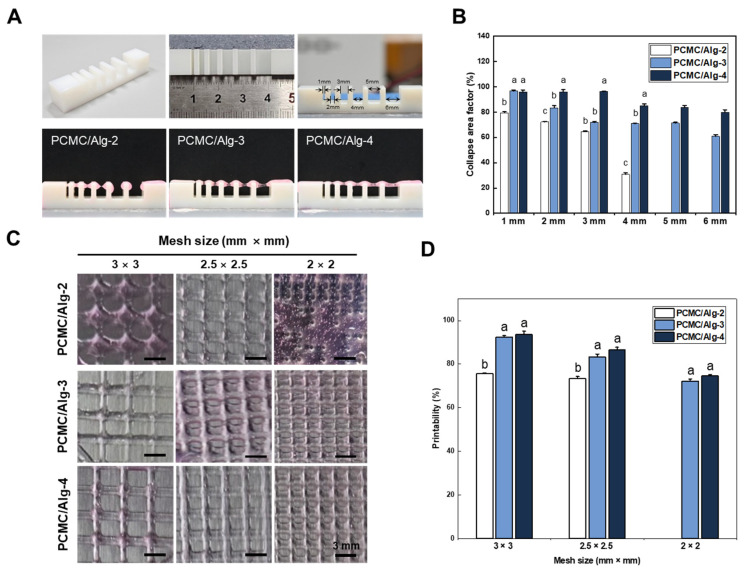
Three-dimensional printability test. (**A**) Filament collapse tests were conducted at various distances between the set pillars. PCMC was subjected to collapse tests using various concentrations of alginate, and the results were analyzed. (**B**) The quantified results of the filament collapse test. The collapse area coefficient decreased with increasing column distance for all compositions. PCMC/Alg-2 did not exhibit overhang capability at distances of 5 mm and 6 mm. In contrast to PCMC/Alg-2, both PCMC/Alg-3 and PCMC/Alg-4 exhibited filament stability without notable collapse. Error bars indicate the standard mean of errors. The same letter means statistical insignificance (Duncan’s new multiple range test, *p* ≤ 0.05). (**C**) Optical images of PCMC/Alg bioink depending on various mesh sizes. Evaluated the capability to print mesh of various sizes on each scaffold. PCMC/Alg-2 failed to print a 2 × 2 mesh size and collapsed, but it could represent larger meshes. PCMC/Alg-3 and PCMC/Alg-4 demonstrated the capability to print mesh structures of all sizes. (**D**) Quantification of 3D printability. Printing ability was calculated as a percentage by dividing the actual mesh size by the intended mesh size. Printability decreased as the mesh size was reduced in all groups. Error bars indicate the standard mean of errors. The same letter means statistical insignificance (Duncan’s new multiple range test, *p* ≤ 0.05).

**Figure 5 polymers-15-04473-f005:**
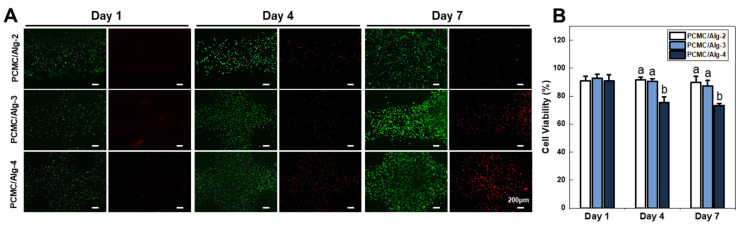
Cell viability test. (**A**) Assessment of C2C12 cell viability in structures printed with PCMC/Alg bioink using Live/Dead (green/red) assay. Live/Dead assays were conducted at predefined intervals: 1 day, 4 days, and 7 days after bioprinting. (**B**) Quantification of cell viability. The percentage was determined by dividing the number of live cells by the total cell count. ImageJ software was used to count live and dead cells from the images acquired through the Live/Dead assay. PCMC/Alg-2 and PCMC/Alg-3 with more porous structures may improve the exchange of growth factors and waste removal, resulting in higher cell viability. They can reduce shear stress when 3D bioprinting because of their lower elastic modulus. Error bars indicate the standard mean of errors. The same letter means statistical insignificance (Duncan’s new multiple range test, *p* ≤ 0.05).

## Data Availability

The data presented in this study are available on request from the corresponding author.

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
