# Peer review of "Development of Plum Seed-Derived Carboxymethylcellulose Bioink for 3D Bioprinting"

_polymers, 2023, doi:10.3390/polym15234473_

Round 1

Reviewer 1 Report

Comments and Suggestions for Authors

1.      In the abstract provide some information about the biocompatibility and antimicrobial behavior of the prepared bio-ink.

2.      Authors should also check the antimicrobial properties of the bio-ink, which is crucial for biomedical/ food applications.

3.      In the conclusion authors should summarize which combination of PCMC and Alginate is the best for 3D printing based on their outputs of cell viability, shearing, collapse etc.

4.      Discussion of results of current study is missing. Authors should discuss their results and should compare with previous studies to draw final conclusion.

5.      Authors should provide the comparison of their results and how their research is different from previous one.

6.      Cite the reference to synthesize PCMC in section 2.3

7.      Consistently use the abbreviations throughout the whole manuscript after first description.

8.      In whole manuscript there are a lot of ambiguous sentences, careful English editing is required.

9.      Provide equation for collapse area coefficient measurement.

10.  Provide error bar for each part of the figure 4C and 5A.

11.  In the conclusion section replace the word ‘persimmon seed’ with ‘plum seed’.

12.  It is highly recommended to cross check manuscript for typos errors, grammar mistakes and sentence structure.

Comments on the Quality of English Language

  It is highly recommended to cross check manuscript for typos errors, grammar mistakes and sentence structure.

In whole manuscript there are a lot of ambiguous sentences, careful English editing is required.

Reviewer 2 Report

Comments and Suggestions for Authors

Please correct the reference part according to the journal instructions (ex: name of the journal either abbreviation or full name; Only the first page or first and last page of the cited article)

The authors need to draw correctly the structure of cellulose and PCMC, because the structure shown in figure 1 is wrong. The authors can use Chemsketch, free software easily downloadable from internet, to draw structures

Extraction of cellulose: this term is inappropriate, since  cellulose is not extracted, it is an enrichment of cellulose from the plum seeds. Talk instead about “cellulose recovery” . Do the authors have any idea of the DP of cellulose, since acidic conditions can decrease this DP?

Characterization of PCMC: What is the DS of PCMC? there is neither mention of a method for determining the DS nor the DS of the synthesized PCMCs. However, this notion is essential  to understand the characteristics of CMC (swelling, solubility, etc.), and can therefore have an influence on the formulation of the ink. This issue must be resolved before considering acceptance of this manuscript.

Please indicate on the graphs whether the results are significantly different or not, as well as the level of significance.

Please correct the reference part according to the journal instructions (ex: name of the journal either abbreviation or full name; Only the first page or first and last page of the cited article)

Comments on the Quality of English Language

Minor editing of English language required

Round 2

Reviewer 2 Report

Comments and Suggestions for Authors

can be published

Author Response

We sincerely appreciate your advice. I acknowledge that your guidance has helped correct the course of our documentation.
